# Retinal Pigment Epithelium Under Oxidative Stress: Chaperoning Autophagy and Beyond

**DOI:** 10.3390/ijms26031193

**Published:** 2025-01-30

**Authors:** Yuliya Markitantova, Vladimir Simirskii

**Affiliations:** Koltzov Institute of Developmental Biology, Russian Academy of Sciences, 119334 Moscow, Russia; simir@mail.ru

**Keywords:** retinal pigment epithelium, phagocytosis, oxidative stress, proteostasis system, mitochondrial dysfunction, ubiquitin–proteasome system, autophagy, heat shock proteins, chaperone mediated autophagy, lysosomes, programmed cell death

## Abstract

The structural and functional integrity of the retinal pigment epithelium (RPE) plays a key role in the normal functioning of the visual system. RPE cells are characterized by an efficient system of photoreceptor outer segment phagocytosis, high metabolic activity, and risk of oxidative damage. RPE dysfunction is a common pathological feature in various retinal diseases. Dysregulation of RPE cell proteostasis and redox homeostasis is accompanied by increased reactive oxygen species generation during the impairment of phagocytosis, lysosomal and mitochondrial failure, and an accumulation of waste lipidic and protein aggregates. They are the inducers of RPE dysfunction and can trigger specific pathways of cell death. Autophagy serves as important mechanism in the endogenous defense system, controlling RPE homeostasis and survival under normal conditions and cellular responses under stress conditions through the degradation of intracellular components. Impairment of the autophagy process itself can result in cell death. In this review, we summarize the classical types of oxidative stress-induced autophagy in the RPE with an emphasis on autophagy mediated by molecular chaperones. Heat shock proteins, which represent hubs connecting the life supporting pathways of RPE cells, play a special role in these mechanisms. Regulation of oxidative stress-counteracting autophagy is an essential strategy for protecting the RPE against pathological damage when preventing retinal degenerative disease progression.

## 1. Introduction

The state of retinal pigment epithelium (RPE) cells plays a central role in the viability and homeostasis of the neural retina. The RPE provides functional support for the retinal neurons and the choroid, thanks to the coordinated components in the endogenous defense system [1,2,3]. Interactions between the RPE and neighboring tissues such as the choroid and photoreceptors maintain the local RPE and vision functions [4,5].

RPE cells and photoreceptors are characterized by high metabolic activity, due to their involvement in the mechanisms of the visual cycle. During the visual cycle and metabolism of rhodopsin, the isomerization of 11-cis-retinal to trans-retinal leads to chain of biochemical reactions and as a result to the production of a large amount of reactive oxygen species (ROS), causing lipid peroxidation (LPO) [6,7]. The high metabolism rate in the RPE and retina induces oxidative stress (OS) in these tissues. The key physiological function of the RPE is associated with protecting photoreceptors from excess light and ROS by means of endogenous defense systems [7,8,9]. The endogenous defense systems of the RPE provide antioxidant protection and redox homeostasis and promote the remodeling of damaged proteins and their repair and/or degradation through autophagy or the ubiquitin–proteasome system [10,11].

Structural, metabolic, and genetic disorders of RPE cells and neighboring tissue structures (the choroid, Bruch’s membrane, and photoreceptors) lead to the accumulation of toxic components, which creates a risk of redox disbalance and OS development [12,13]. RPE cell defects and dysfunction are associated with disorders in the key components of the endogenous defense system in the RPE and contribute to the development of retinal diseases, such as age-related macular degeneration (AMD) and *proliferative vitreoretinopathy* (PVR) [14,15,16].

Maintaining the structural and functional integrity and homeostatic balance in RPE cells constitutes an urgent problem in biomedicine and ophthalmology. The regulatory network of the endogenous defense system defines the physiological functions of the RPE cell layer, ensuring retinal neuron viability and homeostasis and enabling adaptation in response to cellular stress [17].

A number of therapeutic approaches are aimed at maintaining the stability of intercellular contacts and the functionality of RPE cells. Thus, the use of factors such as nicotinamide or lysophosphatidic acid contributes to the formation of the tight junctions in the RPE [18,19]. Several proposed therapeutic strategies targeting RPE cells are aimed at activating the antioxidant defense and alleviating both OS and mitochondrial and lysosomal dysfunction, which are the main mechanisms of RPE dysfunction resulting in AMD. Photobiomodulation agents can also remove drusen in AMD by activating RPE phagocytosis [15,20,21,22].

Regulation of RPE cell dysfunction aimed at preventing RPE-related neurodegenerative pathologies is a priority for modern research. Potential therapeutic strategies involve targeting autophagy as an important biological process to maintain homeostasis in RPE cells [23,24,25].

In this review, we have performed an analysis of the current state of the research focusing on the endogenous antioxidant defense system in the RPE with a focus on autophagy and molecular chaperones, highlighting them as potential targets as therapies for RPE-related cell death and retinal degenerative pathologies. The molecular changes and signaling pathways involving heat shock proteins (HSPs) that control the RPE response to injury remain largely undefined. Given the important roles of HSPs in the RPE in retinal disease and health, we reviewed recent data on their roles in the molecular protection of RPE cells obtained using modern molecular genetic methods. It would be of interest to fully characterize their roles in the molecular defense responses of RPE cells to injury as a result of trauma or disease affecting the RPE and the retina. Autophagy is a key example of such molecular defense responses.

## 2. RPE Functions for Maintaining the Viability of Photoreceptors

RPE cells form a monolayer of polygonal pigmented cells. RPE cells are located between the choroidal vasculature of the retina and the outer segments of retinal photoreceptors. RPE cells play various roles in the retina and the choroid, maintaining the homeostasis of these tissues [17]. On the basal side, RPE cells interact with Bruch’s membrane, which is closely associated with the choroid. On the apical side, RPE cells interdigitate with photoreceptor outer segments (POS), forming tight intercellular contacts [17,26,27].

Numerous vital RPE functions include (1) the utilization of shed POS discs (by phagocytosis); (2) maintaining the visual cycle in retina (by degrading or recycling components of the visual cascade such as visual chromophore 11-cis-retinal); (3) the absorption of light energy (by melanosome bleaching); (4) the maintenance of redox homeostasis (by free radicals and ROS scavenging as well as the activities of antioxidant enzymes and chaperone systems); (5) the blood–retinal barrier (through the formation of Bruch’s membrane and tight junctions); and (6) the transport of nutrients and metabolites to the photoreceptors and choroid (through secretion, pinocytosis, endocytosis, and exocytosis) [17,20,28,29,30,31,32].

RPE cells constantly renew the “exhausted” photoreceptor discs by phagocytosis of the used POS, which are subjected to autophagic and lysosomal degradation. POS discs are enriched in ROS and lipofuscin (LF), which is the main product of LPO [9]. Superoxide radicals generated by LF under the action of visible light have a destructive effect on RPE melanosomes, which perform screening and antioxidant functions [33].

A high metabolism rate of RPE cells, high oxygen consumption and intensity of energy metabolism processes, a constant renewal of POS membrane discs using phagocytosis, high levels of polyunsaturated fatty acids and exposure to light are the main RPE characteristics needed to maintain a normal physiological state of the photoreceptors and the retinal function [1]. On the apical side of RPE cells, abundant melanosomes, which absorb stray light during the visual function, are distinguished from the nucleus and melano-LF granules on the basal side [34,35,36].

Intense mitochondrial metabolism, the phagocytosis of POS discs, the phototoxic activity of LF, and the photosensitization of hemoglobin precursors are the main sources of ROS, both in the RPE and photoreceptors [12]. In differentiated RPE cells, redox homeostasis largely depends on autophagic clearance and the intensity of the accumulation of cellular debris, including LF. With age, the intracellular redox balance shifts toward the intensification of oxidative reactions, which enhances the production of ROS [37].

## 3. Major Intercellular and Molecular Events That Contribute to Oxidative Damage in RPE Cells

In retinal cells, ROS are produced by lysosomes (phagosomes), peroxisomes, melanosomes, and intracellular membrane-bound NADPH oxidases [38,39].

Disruption of RPE polarity as a result of a disrupted interactions with the photoreceptors is accompanied by a decrease in intercellular adhesion, reduced barrier function, increased ROS production, migration of RPE cells, and retinal detachment, leading to several RPE-related diseases, such as PVR and AMD [40,41,42] (Figure 1).

The destructive influence of ROS manifests itself in the oxidation of proteins and membrane lipids and can result in DNA damage [43]. Excessive intracellular ROS production leads to the loss of the inherent structure of some proteins and to their aggregation. This can contribute to endoplasmic reticulum (ER) stress and cause a response to unstructured proteins. The accumulation of aggregated proteins increases OS, leading to lysosome damage and inflammation [44,45].

The structural disorders and dysfunction of RPE cells impair photoreceptor metabolism and are caused by an excessive accumulation of LF, releasing lysosomal-degrading enzymes into the cytosol, which can lead to cell death [46,47]. The process of LF formation is accompanied by an inhibition of proteasome activity in the RPE [48]. In turn, a decrease in proteasome activity is accompanied by the accumulation of LF in the RPE and a decrease in the intensity of autophagy [10,49].

ROS overproduction in the mitochondria contributes to ROS production and can result in mitochondrial DNA damage [50]. Under OS, the mitochondria may aggravate the production of ROS, which can lead to cell apoptosis [37].

RPE cells have abundant mitochondria to provide for the energy needs of the outer retina cells [51]. Damaged respiration due to mitochondrial dysfunction in RPE cells is one of the main links to AMD pathogenesis [22,52]. ROS levels can increase as a result of interactions between the mitochondria and cytochrome C oxidase [12,53].

Exposure of LF from melano-LF granules to light manifests itself in the production of ROS, which destroys the melanin component. With age and the development of degenerative diseases of the RPE and photoreceptors, a large number of melano-LF granules are formed. Excessive accumulation of melano-LF granules leads to a decrease in and even disappearance of melanin in pigmented RPE cells, thereby reducing the protection of the cells from ROS [34].

It has been shown that impaired lysosomal degradation owing to the accumulation of LF is closely related to autophagy disorders in AMD. Activation of lysosomal proteases from the cathepsin class in the RPE promotes the destruction of POS, as well as the formation of end products of LPO and oxidized low-density lipoproteins [54]. Their accumulation leads to the excessive accumulation of LF. In turn, excessive intracellular deposition of LF enhances OS and the formation of autolysosomes and drusen in the RPE [55]. The impaired lysosomal degradation due to the accumulation of LF is closely related to autophagy disorders and apoptosis, as shown during AMD. Inflammasome priming during inflammation development in retinal pigment epithelial cells increases their susceptibility to LF phototoxicity, thereby switching from the cell death mechanism via apoptosis to the pyroptosis pathway [54,56].

## 4. Endogenous Defense System in RPE Activated in Response to OS

In response to OS conditions, the intensity of phagocytosis in RPE cells increases. This also manifests itself in the increased phagocytosis of the apoptotic fragments of dying cells and the utilization of Bruch’s membrane metabolic products [9].

The endogenous defense systems of RPE cells include the antioxidant system, different kinds of autophagy, and a multilevel system of chaperones. Antioxidant enzymes (superoxide dismutase [SOD], catalase, and cytochrome P450 monooxygenase) and non-enzymatic molecules (thioredoxin, glutathione ascorbate, and β-carotene) [57] take part in maintaining redox homeostasis. Among them, SOD2 is a primary enzyme activated in response to OS and protects the cells from damage by removing ROS and maintains the function of mitochondria [53]. Under normal conditions, these systems maintain cellular homeostasis, but they can activate any kind of programmed cell death under stress conditions (Figure 2).

RPE pigments absorb light, protecting the retina against oxidative damage, continuous exposure to light, and the photo-oxidation of LF [58]. Melanin is a key component of the antioxidant system. The density of melanin is relatively high in the RPE cells adjacent to the central zone of the retina, while the highest density is found in the fovea [59]. A significant age-related decrease in the amount of melanin, as well as in the number of melanosomes that act as screening light filters and antioxidants in RPE cells, may lead to an increase in the risk of oxidative and photo-OS in the structures of the eye [34]. Melanin decreases the photo-oxidation of LF by shielding from the harmful bright light in the RPE [60] and removes ROS, thus contributing to protection from OS [35].

Autophagy is the key biological process necessary for maintaining cellular homeostasis by lysosomal degradation of unused and damaged cellular components [61]. In RPE cells, autophagy is closely related to the regulation of proteostasis and redox homeostasis [62].

## 5. Chaperone Defense System of Heat Shock Proteins in the RPE

Chaperone proteins play a critical role in all taxonomic groups and are involved in ensuring both the folding of newly synthesized peptides into their mature conformation, the refolding of misfolded proteins, and the movement of proteins to subcellular compartments [63]. They function under normal conditions and can be activated in response to internal and external stressors.

The most common and studied group of chaperones are HSPs. They can be categorized into several families based on their molecular size, structure, and function: small HSPs (HSPB), HSP40 (DNAJ), HSP60/HSP10 (HSPD/E), HSP70 (HSPA), HSP90 (HSPC), and HSP100/110 (HSPH) [64,65]. To date, up to 83 HSP genes have been identified in the human genome: 4 for HSP110, 4 for HSP90, 13 for HSP70, 1 for HSP60, 50 for HSP40, and 11 for small HSPs [65].

RPE cells are exposed to chronic OS from three sources: high levels of oxygen consumption, LPO from phagocytized POS, and light stimuli. Cells constitutively express many types of HSPs but increase their expression to stabilize and restore cellular homeostasis under stress (OS, hypoxia, high glucose, and others).

HSPs have been demonstrated to play a key role in cellular responses to stress, serving as an adaptive response center and stabilizing cellular structures [66]. Many HSPs are upregulated during cellular stress and are expressed with high tissue specificity. There is a strong connection between the stress regulation systems in cells and the mechanisms that control their growth and differentiation. HSPs not only act as molecular chaperones, but also participate in cell homeostasis and cell viability, as well as cell differentiation processes. The ability of HSPs to protect cells is a key aspect of their functioning to support cell viability [67,68].

αA-crystallin contributes to the activation of phosphorylation reactions in the PI3K/Akt signaling pathway, which ensures the resistance of RPE cells to the action of OS [69]. In the RPE, αB-crystallin protects proteins against aggregation and unfolding and stimulates the production of vascular endothelial growth factor (VEGF) in response to inflammation, which contributes to neovascularization in AMD [70].

In the RPE, OS induces the activation of redox-dependent antioxidants, DJ-1 chaperones and alpha-1 microglobulin. The latter can directly bind and neutralize ROS [71,72].

Almost all the proteins of the HSP family, with the exception of sHSPs, need ATP (adenosine 5ʹ-triphosphate) hydrolysis for their activity. The interactions of Hsp60 with the co-chaperone Hsp10 control the substrate-binding and ATPase activities of Hsp60 [73]. Despite the fact that the Hsp60 protein is localized mainly in mitochondria, it can, like other stress proteins, change its intracellular localization in response to OS and is expressed on the cell membrane [74].

Each member of the HSP70 family can respond specifically in cells and tissues, depending on the type of stress [67]. For example, the human Hsp70B protein is strictly expressed in response to heat shock. This distinguishes Hsp70B from Hsp70, which is characterized by a basal level of expression and activation under different stress conditions [75].

HSP110 is involved in the proteolysis of α-synuclein and/or the prevention of associated neurodegeneration in mammals. The Hsp110 protein has a high structural and functional similarity to Hsp70 and performs cytoprotective functions against the negative effects of OS. It is assumed that this function of HSP110 as a co-chaperone is associated with the regulation of HSP70 expression [76].

Hsp90 proteins control the quality of proteins, ensuring their transport to the proteasomes for destruction in cases of protein defects that prevent them from performing their functions. Normally, Hsp90 protein expression is insignificant in the total content of all proteins in the cells, but its amount increases during stress [77]. During heat shock, HSP90 activity is necessary to restore the functions of misfolded and denatured proteins. The synthesis of Hsp90 also increases under the influence of other stress factors such as inflammation, which cause OS [77].

Neurodegenerative pathologies are associated with dysfunction of sHSPs [78]. Disorders in the expression of HSPs are also associated with inflammatory reactions, as well as with the death of pigment epithelial cells and retinal neurons in various eye pathologies [79,80]. A decrease in the amount of HSP proteins (HSP60, HSP70, and αA-crystallin) occurs against the background of increased OS in the human retina in AMD [81]. Intracellular aggregates formed by incorrectly folded proteins can form β-amyloid structures as a result of dysfunction of sHSPs. These structures are characterized by their high stability to proteolytic cleavage, leading to their excessive accumulation, which causes cell death. sHSPs are actively involved in the formation of elaborate protein complexes that can affect the activity of cytoskeletal proteins. It is noted that in Parkinson’s disease, Lewy bodies consisting of sHSPs and neurofilaments are found in a complex of protein aggregates in the neurons, but the mechanisms of the accumulation of these complexes have not been clarified [82]. HSP27 also plays an important role in the organization of microfilaments. It interacts with actin filaments, is involved in programmed cell death (apoptosis), and is important for cell survival under stress conditions [83,84].

In general, HSPs (HSP70, HSP90 and others) primarily act as molecular chaperones, preventing cell damage from proteins with disrupted conformation caused by OS. HSPs also participate in the mechanisms of redox balance regulation in cells. In a recent study, a polymorphic variant of HSP70 was discovered, namely, HSP70-2, which is a sensor of the redox balance of cells and changes in OS. The activity of HSP70-2 is associated with the concentration of ROS in the cell, which plays a role in various functional disorders [85].

## 6. Proteolytic and Autophagy Defense Systems

As a result of LPO, which is rich in POS, and the metabolism of trans-retinol, RPE cells produce excess amounts of ROS, which initiate OS. An additional source of ROS is constituted by the products of photo-oxidation of LF [6,36]. RPE cells have numerous antioxidant systems that ensure the removal of excess ROS and the restoration of redox homeostasis [20,59]. One of the most universal systems for maintaining and restoring proteostasis is the chaperone system. The most studied chaperones are HSPs that provide a unique system to regulate the traffic of newly synthetized proteins between cellular compartments, to promote the refolding of misfolded proteins, and to inhibit the formation of toxic protein aggregates [86,87]. Among the defense mechanisms responsible for maintaining cellular homeostasis, HSPs provide the only way of restoring the configuration of unfolded or misfolded proteins [88]. Once the capacity of HSPs is exceeded, misfolded, aggregated, and damaged proteins, as well as damaged organelles, undergo degradation using proteolytic systems. Four different well-coordinated systems are responsible for protein degradation in eukaryotic cells: (1) the proteasome-based UPS, which degrades most long- and short-lived normal and abnormal intracellular proteins; (2) mitochondrial proteases, which degrade mitochondrial proteins; (3) calcium-activated calpains, which degrade membrane and cytoskeletal proteins and several membrane-associated enzymes; and (4) lysosome-based autophagy, which degrades cellular organelles, membrane proteins, and endocytosed proteins [89,90]. Both autophagy and proteasomal clearance are especially crucial for cell types with no proliferation, such as RPE cells. These systems regulate cellular homeostasis, control the quality of mitochondria and the production of ROS under conditions of normal functioning, and provide antioxidant protection to RPE cells under OS conditions [91,92].

Proteasomes are efficient in the degradation of small and short-lived proteins, whereas larger and longer-lived substrates (including lipids) are degraded by autophagy [93]. Despite the unique role of proteasomes in the regulation of cellular homeostasis, they are functionally related and can act together, which is especially pronounced under OS conditions. Proteasome inhibition may be compensated by increased autophagy [94].

In both systems, molecular chaperones play important roles in the recognition and selection of the proteins or organelles to be degraded [64,95,96].

### 6.1. Ubiquitin–Proteasome System (UPS)

UPS-mediated protein degradation is a multistep process involving various proteins. In particular, a protein to be degraded is first labeled with ubiquitin and then recognized and degraded by a proteasome [97]. Proteasomes are multi-protein complexes responsible for the selective degradation of misfolded and high-turnover proteins in the cytoplasm. These proteins have many proteolytically active sites that break down proteins into peptides. The 26S proteasome is highly conserved throughout eukaryotes, where it is found in both the nucleus and cytoplasm [98]. The proteasome is composed of 33 subunits assembled in two sub-complexes, as well as the 20S core particle, which bears the actual protease active sites and is flanked at one or both ends by the 19S regulatory particle to form the 26S proteasome [99]. Proteasome assembly requires the assistance of proteasome assembly chaperones. In mammals, four evolutionarily conserved 19S regulatory particle assembly chaperones (PACs), including p27, p28, S5b, and Rpn14/PAAF1, are needed for regulatory particle assembly [98,100]. Formation of the proteasomal 20S core complex relies on the function of the other five chaperones PAC1-PAC4 (proteasome assembly chaperones 1–4) and POMP (proteasome maturation protein) [101,102].

Involvement of αB-crystallin in protein degradation pathways has also been discovered. αB-Crystallin interaction with C8/a7, one of the 14 subunits of the 20S proteasome, has been reported both in vitro and in vivo. This interaction is highly specific since C8/a7 does not bind to αA-crystallin or HSP27 [103]. A mutation in αB-crystallin decreased Atg7 expression, which is a mediator of autophagosomal biogenesis [104]. αB-crystallin can also promote the degradation of certain misfolded proteins that cannot be converted to their native state after repeated cycling through the chaperone systems by the proteasome [105,106]. The opposite role is played by HSP90, which interacts with and protects the transcription factor SP1 from degradation in the ubiquitin–proteasome pathway [107].

Degradation of a protein by the UPS is typically mediated by ubiquitination of the target protein, which involves the energy-dependent covalent attachment of the small protein ubiquitin (Ub) to one or more lysines within the protein via the concerted action of three substrate-specific E1-E3 Ub-protein ligases [98]. Normally, the cell maintains a balance between the formation of ROS and their neutralization [108]. Disruption of redox homeostasis can lead to OS and the formation of oxidation products of proteins, lipids, and other macromolecules [109]. Proteasomal degradation of oxidized proteins mainly occurs in the cytosol, which is zone of greatest production of oxidized proteins. On the other hand, the nucleus is well protected against the formation of oxidized proteins and their accumulation, as a result the high proteasome content [110]. Among these proteins, only slightly oxidized forms of proteins re suitable substrates for the proteasome. Highly oxidized proteins are likely to exist in stable aggregates due to covalent cross-links, disulfide bonds, or hydrophobic interactions. Such proteins aggregate, together with oxidized lipids, thereby forming LF granules. They are no longer suitable as a substrate for proteasomes and are degraded by lysosomes [111]. Human AMD donor RPE in which OS is activated exhibited a significantly higher content of the proteasome as well as HSP27 and HSP90 [112].

### 6.2. Phagocytosis

POS discs are constantly shed and subsequently phagocytosed by the RPE before new outer segments are constructed at the cilium. RPE cells are the most active phagocytic cells in the human body, and defects in the phagocytic process lead to impaired retinal function [113].

The process of POS phagocytosis is triggered by ligand secretion by the RPE, which provides specific binding of the POS discs to the RPE apical membrane. One of these ligands, milk fat globule-EGF8 (MFGE8), binds to exposed phosphatidylserines on POS fragments, bridging them to aVb5 integrin receptors on the apical surface of the RPE [114]. This binding initiates two downstream signaling cascades. On the one hand, it stimulates Mer receptor tyrosine kinase (MERTK) via aVb5 integrin-associated focal adhesion kinase [115]. On the other hand, it activates RAC1-GTPase [116], which leads to F-actin recruitment to the phagocytic cup.

The rearrangement of cytoskeletal filamentous actin RPE (F-actin) is required for POS internalization [117]. The phagocytic cup, formed by the ordered aggregation of F-actin in combination with POS, is the key to the initiation of phagocytosis [118]. Cyclic TCP-1 complex, also known as chaperonin-containing TCP-1 (CCT), consists of eight parallel subunits (CCT1–8) [119]. A recent study showed that CCT is required for efficient assembly of actin myofilaments [120], and CCT5-specific ATP binding is required for efficient actin folding [121]. In addition, CCT5 controls lysosome biogenesis through the actin cytoskeleton [122]. The association of HSP90 with F-actin, but not with α-tubulin, is important for phagosome formation. Silencing of HSP90 (siHSP90) reduced expression of cytoskeletal proteins and the phagosome marker (Rab5) and successfully diminished phagocytosis in U937-derived macrophages [123].

Mesoderm development candidate 2 (Mesd or Mesdc2) has been identified as facilitating phagocytosis in the RPE. Mesd is an ER-located chaperone that facilitates the folding of the low-density lipoprotein receptor-related protein (LRP) family [124]. Mesd is predominantly expressed in POS fragments, which are an ER-free cellular compartment [125]. It has been demonstrated that Mesd can be released from shedding POS fragments and stimulates their phagocytosis by RPE cells binding to phagosomes [126].

It is known that most lysosomal RPE enzymes function within a narrow pH range in the acidic environment of the lysosomal lumen [127]. The acidic luminal environment is primarily created by the vacuolar-type H+-ATPase (V-ATPase), which pumps protons into the lumen [128]. In RPE cells, CRYBA1/βA3/A1-crystallin is found in lysosomes, where it functions as a regulator of endolysosomal acidification by modulating V-ATPase, to control of both phagocytosis and autophagy. CRYBA/βA3/A1-crystallin 1 directly influences lysosomal V-ATPase activity. Crosstalk between V-ATPase and mTORC1 is required for the regulation of autophagy, and CRYBA1 regulates the activity of the mTORC1 signaling pathway [129].

### 6.3. LC3-Associated Phagocytosis

A new type of phagocytosis has been described in the RPE, where the maturing phagosome acquires microtubule-associated protein 1 light chain 3B (LC3B) classically associated with macroautophagy. This hybrid autophagy–phagocytosis degradation pathway is termed LC3-associated phagocytosis (LAP) [130,131,132]. Microtubule-associated protein 1 light chain 3B (LC3B) mediates the physical interactions between microtubules and components of the cytoskeleton. LAP and macroautophagy both rely on the lipidation of LC3 to LC3II, which ensures its membrane link with double-membrane autophagosomes in autophagy or single-membrane phagosomes in LAP. Lipidation of LC3 involves a complex of Atg5-12 and Atg16L; these factors act in cooperation as an E3-like enzyme to transfer phosphatidylethanolamine (PE) to LC3 [133]. The lipidated form of the autophagy protein LC3 binds to the phagosome in a manner dependent on Atg5 and Beclin1, but independent of the autophagy preinitiation complex including the Ulk1/Atg13/Fip200 [134]. The composition of the PI3K complex consisting of subunits involved in LAP also differs from that of autophagy. The LAP complex contains the RUN domain protein Rubicon, while the autophagy complex includes Atg14 [131]. Phagosomes require Atg5 to move through the RPE and enter the lysosomal compartment for degradation. In addition to the degradation of shed POS discs, this noncanonical form of autophagy supports optimal visual function by supplying a portion of the retinoids required for chromophore synthesis [132].

LAP coexists in the RPE with canonical phagocytosis [132]. New adapter proteins have been found that bind LC3, targeting specific organelles to lysosomes. One of them is melanoregulin, which is specific for phagosomes [130]. Notably, only part of the POS-phagosome is degraded by LAP. A significant portion of POS phagosomes are ubiquitinated and, as a result, can be included in phagosomes that have LC3B, melanoregulin, or SQSTM1 as receptors.

In RPE cells, phagocytosis starts with the capture and ensheathment of POS by the apical processes of the RPE. Ensheathment is stimulated by MERTK ligands, GAS6 and PROS1, but not by V5 integrin receptor ligands, MFG-E8 and vitronectin. Remarkably, the ensheathment participates in POS fragmentation before their internalization. It is suggested that MERTK activation is required for ensheathment-mediated POS fragmentation before internalization [135].

In the RPE, LAP plays a necessary role in cell homeostasis through the optimal clearance of phagocytosed POS fragments. LAP supports (1) the visual cycle by recycling retinoids, (2) retinal metabolism (e.g., by metabolizing lipids to generate ketones), (3) lipid homeostasis, and (4) the synthesis of anti-inflammatory lipids [132,136,137].

RPE cells use Rubicon (RUN domain and cysteine-rich domain containing Beclin 1-interacting protein) similarly to macrophages to stimulate LAP and efficiently degrade phagocytosed cargo. It was found that Rubicon expression is highest in the morning during the time of maximal POS degradation. The phagocytosis of outer segments activates EGFR (epidermal growth factor receptor), suppressing autophagy. When exposed to starvation stress, RPE cells activate autophagy, which impairs phagocytic degradation. Thus, RPE cells maintain a balance between phagocytosis and autophagy, ensuring their long-term functions and retinal homeostasis [138]. ARPE-19 cells are efficient at phagocytizing rod POS under both normal and high-glucose conditions. However, under high-glucose conditions, ARPE-19 cells treated with oxidized rod POS fragments accumulated malondialdehyde and LF and displayed altered LC3, GRP78, and caspase-8 expression compared with untreated and unoxidized rod POS-treated cells [139]. Thus, phagocytizing cells grown in high glucose appear more prone to suffer a permanent oxidative insult than those grown under normal conditions.

### 6.4. Autophagy

Autophagy is the catabolism of intracellular material in specialized cellular structures, namely, lysosomes. Autophagy is involved in maintaining cellular homeostasis, adaptation to stress, immune responses, and regulation of inflammatory processes.

In mammals, three main autophagic pathways target substrates for lysosomal degradation. Macroautophagy involves the formation of autophagosomes, double-membrane vesicles, to transport substrates to lysosomes for degradation. In chaperone-mediated autophagy, lysosomal degradation is regulated by the interaction between lysosomal receptor LAMP2 (lysosome-associated membrane protein 2) and chaperone Hsc70, facilitating the selective access of substrates with the target motif to the lysosome. In microautophagy, lysosomal degradation is the simplest, involving the direct invagination and sequestration of substrates into the lysosome [140].

#### 6.4.1. Macroautophagy

Protein aggregates, cellular organelles, and protein complexes in signaling cascades are typically degraded through selective autophagy. Selective autophagy uses autophagy receptors that bind to specific structural elements of the protein or organelle that must be eliminated [141]. Ubiquitin modification is also part of the signal that marks cellular components for destruction. Autophagy receptors, such as SQSTM1/p62, NBR1, and optineurin, contain domains that recognize ubiquitin and unfolded protein structural elements in cargo [142,143,144]. After receptor binding, the cargo is chaperoned to the autophagosome where it is degraded. The autophagosome is decorated by ATG8 (autophagy-associated protein 8)/LC3 (microtubule-associated protein 1A/1B-light chain 3), which are recognized by the LIR (LC3-interacting regions) motif that is present on all autophagy receptors [30]. The RPE phagocytizes and degrades the POS in autophagosomes. The process of autophagosome formation includes initiation and elongation phases, during which an insulating membrane is formed. This membrane grows during the nucleation phase to form the mature autophagosome. Fusion of the autophagosome with lysosomes will allow the engulfed material to be degraded during the degradation phase [145]. All stages of the process from induction to autophagosome formation and its fusion with the lysosome are regulated by members of the ATG family of proteins. More than 35 ATG genes have been identified that control the autophagy process [146].

The RPE maintains basal autophagy for cellular homeostasis, with autophagic variations common in aging and diseased cells [55,62,147]. Hypoxia, OS, the unfolded protein response, and inflammation are typical inducers of autophagy [11,147,148,149]. These conditions are also associated with RPE aging and AMD. In the RPE, p62 promotes autophagy and simultaneously enhances an Nrf2-mediated antioxidant response to protect against acute OS [150]. Members of the Bcl-2-associated anthanogen (BAGs) may be also cytoprotective by inducing autophagy [151].

#### 6.4.2. Chaperone-Mediated Autophagy

Chaperone-mediated autophagy (CMA) degrades the proteins that have a specific tag related to pentapeptide KFERQ [152]. These proteins first bind to cytosolic chaperone hsc70 and its co-chaperones to unfold, then bind to lysosomal membrane protein LAMP-2A, and are directly translocated across the lysosomal membrane, without requiring the formation of intermediate vesicles or membrane deformation [153,154].

Molecular chaperones (Hsp70, Hsp40, and Hsp90) can form a complex with their transcription factor HSF1 in the cytosol. Upon binding misfolded proteins Hsp70, Hsp40, and Hsp90 dissociate from HSF1, which is activated via phosphorylation, and are trafficked to the nucleus to increase chaperone expression. The substrate proteins in the cytosol are selectively bound by the KFERQ-motif with chaperone proteins and transported to the lysosomal lumen via CMA. In the lysosome, the substrate proteins are digested and degraded by lysosomal enzymes. The KFERQ-motif is a conserved peptide sequence in the target proteins recognized by cytosolic Hsc70 and chaperoned into the lysosome via the LAMP-2A channel. LAMP-2A is the key component of chaperone-mediated autophagy. LAMP-2A organizes into specific protein complexes at the lysosomal membrane. HSP90 regulates the stability of lysosome-associated membrane protein-2a (Lamp-2a) in the process of CMA. The central regulator of this degradation pathway is constitutively expressed heat shock protein 70 (HSC70). It recognizes the KFERQ motif in protein sequences and stimulates protein translocation across membranes [155]. There is a group of chaperone proteins that interact with HSC70 and regulate its activity or may themselves act as chaperones. They include heat shock protein 40 (HSP40), which stimulates the ATPase activity of HSC70; HSC70-interacting protein (HIP), which stimulates complex assembly; heat shock protein 90 (HSP90), which can refold unfolded proteins and/or prevents the degradation of unfolded proteins from aggregating; and organizing protein HSC70–HSP90 (HOP), which binds the HSC70–HSP90 chaperones [156,157]. The chaperone complex is associated with the target protein and LAMP-2A on the cytosolic surface of the lysosomal membrane. The main function of chaperones is unfolding the target protein before its transport through the lysosomal membrane [158]. Lys-hsc70 induces disassembly of LAMP-2A from the 700 kDa complex once the substrate has crossed the membrane, and lys-hsp90 stabilizes LAMP-2A during its transition from monomeric to multimeric forms [159].

The main integral transmembrane protein of lysosomes, LAMP-2, is represented by three isoforms. All of them have a short C-terminal cytoplasmic domain (11 amino acids), one transmembrane domain, and a large, highly glycosylated luminal domain. LAMP-2A, B, and C, share an identical lumenal region, but differ in their trans-membrane and cytosolic domain [160,161]. Only the cytosolic domain of LAMP-2A is recognized by the CMA substrates [162]. Organization of LAMP-2A into multimeric complexes, required for CMA activation, only occurs outside the cholesterol-rich microdomains of the lysosomal membrane, whereas the LAMP-2A located within these regions is susceptible to proteolytic cleavage and degradation [163].

CMA is activated under OS, prolonged nutrient deprivation, or exposure to toxic compounds that induce protein damage [164,165]. The first CMA regulation point is constituted by the target proteins that become more accessible after oxidation [166]. This is at least partly due to the fact that oxidized proteins are more easily unfolded, which is necessary for their translocation across the lysosomal membrane [167]. Post-translation modifications on substrate proteins not only generate a KFERQ-like motif, but also change the state of exposure of the KFERQ-like motif so that Hsc70 can recognize or de-identify the substrate proteins [168]. Another CMA regulation point is LAMP2A. Oxidation makes lysosomes more active in CMA due to increased LAMP-2A in the lysosomal membrane and HSC70 in the lysosomal lumen [166]. It is LAMP-2A that regulates CMA activity [165,169]. A portion of full-sized LAMP-2A resides within the lysosomal lumen, perhaps complexed with lipids. These molecules are able to reinsert into the lysosomal membrane when CMA is activated [170]. In addition, the degradation rate of LAMP-2A can be regulated through the activity of lysosomal cathepsin A [169]. Lys-Hsp90 plays a role in maintaining LAMP2A stability, while Lys-Hsc70 induces LAMP2A to disassemble for a new cycle [159]. In rainbow trout, activation of CMA upon high-glucose exposure was mediated by generation of mitochondrial ROS and involved the antioxidant transcription factor Nrf2 [171]. It was demonstrated that the signaling mediated by P2X7, a member of the purinergic family of receptors, can increase Hsc70 and LAMP2A mRNA levels, allowing LAMP2A association with the lysosomal membrane under inflammation conditions [172]. In humans and mice, NRF2 performs a positive regulatory function for CMA through binding to the regulatory elements in the LAMP2 gene and enhancing its transcription [173]. However, signaling mediated by the RARα (retinoic acid receptor-α) has been shown to negatively regulate the transcription of CMA components, including LAMP2A [174].

In addition, the lysosome-associated form of GFAP (glial fibrillary acidic protein) has been shown to stabilize the multimeric translocation complex in response to starvation or OS. In this case, GTP mediates the release of EF1α from the lysosomal membrane, promoting the self-association of GFAP and disassembly of the CMA translocation complex, thereby reducing CMA [175].

Moreover, lysosomal Akt and kinase TORC2 (target of rapamycin complex 2) regulate the activity of CMA by controlling the dynamics of assembly and disassembly of the CMA translocation complex on the lysosomal membrane [176].

#### 6.4.3. Microautophagy

Microautophagy is a catabolic process, in which the dysfunctional or superfluous proteins and organelles are delivered directly to the endosomal or lysosomal lumen [177,178]. Microautophagy is classified as non-selective bulk degradation, which occurs under conditions of starvation and provides essential nutrients for cell survival [141]. This process largely depends on the endosomal sorting complexes required for transport (ESCRT) I and III systems and the protein chaperone, hsc70 [179].

Only microER-phagy has been identified in mammalian systems [180]. Mammalian microautophagy pathways all direct their cargo to late endosomes/multivesicular bodies (LEs/MVBs) [181]. The degradation of cytosolic proteins in LEs/MVBs is referred to endosomal microautophagy. As in the case of CMA, endosomal microautophagy may require recognition of a KFERQ-like motif by HSC70 [182]. HSC70 binds to phosphatidylserine in LE/MVB membranes, triggering substrate internalization into the lumen via membrane invaginations that form in an ESCRT-dependent manner. Cargo protein degradation can occur in the LE/MVB compartment itself, although the bulk of degradation occurs after LE/MVB-lysosome fusion [183].

## 7. Programmed Cell Death and Chaperones

The regulation of cell death and survival is under strict control in eukaryotic development and tissue homeostasis. It is distinguished between accidental and programmed cell death (PCD). PCD is mediated genetically and is tightly controlled [184,185]. Cell death can be induced by the genetically programmed suicide mechanisms of apoptosis, necroptosis, and pyroptosis, or it can be a consequence of dysregulated metabolism, as in ferroptosis [186,187].

The death of RPE and other eye tissues occurs through apoptosis, necrosis, pyroptosis, and ferroptosis pathways (Table 1) under stress conditions, with aging, or as a result of pathological processes [188,189,190,191].

### 7.1. Apoptosis

The development of apoptosis is associated with an inhibition of growth and division, leading to controlled death without a leakage of its contents into the environment. Apoptosis is initiated by the activation of a chain of caspases, which belong to the class of cysteine-aspartic proteases [196].

Cell damage activates initiator caspases (caspases 8 and 9), which trigger the activation of effector caspases (caspases 3, 6, and 7). Apoptosis leads to DNA and nuclear fragmentation, cytoskeletal destruction, and the formation of apoptotic bodies. This process can be triggered through intrinsic and extrinsic signaling pathways. The intrinsic pathway, which is OS dependent, is associated with mitochondrial dysfunction and depends on factors secreted by mitochondria [196]. The extrinsic apoptosis pathway is activated after binding of the TNF1 to Fas-associated death receptors. The chain of reactions then activates caspase-8, which induces caspase-3-based apoptosis [197]. Bax translocation to mitochondria is mediated by signals from c-Jun N-terminal kinases and p38 mitogen-activated protein kinase in OS [198,199].

RPE cells constitutively express members of the sHSP family αA- and αB-crystallin, which can function as anti-apoptotic proteins induced during OS [200,201]. Under OS conditions, the antioxidant activity of chaperone proteins αA and αB crystallins increases, which prevents OS-mediated apoptotic cell death of RPE cells. In the RPE cells of mice with the αA-crystallin or αB-crystallin gene knocked out, the accumulation of ROS was followed by the degeneration of retinal photoreceptors under conditions of experimentally simulated OS [202,203].

It is proposed that αB-crystallin may inhibit apoptosis through interaction with p53, preventing its translocation to mitochondria and blocking the apoptotic signaling [204]. p53 is involved in the initiation of the calcium-activated RAF/MEK/ERK signaling pathway of apoptosis, which can be suppressed by αB-crystallin via inhibition of RAS activation [201]. αB-Crystallin interacts directly with the pro-apoptotic members Bax and Bcl-XS and P53 polypeptides in vitro and in vivo, with sequestration of these proteins preventing the translocation to mitochondria and hence suppressing apoptosis [205,206]. Exogenously added recombinant human αB-crystallin was taken up by stressed cells and protected these cells from apoptosis by inhibiting caspase-3 and through poly (ADP-ribose) polymerase activation [207]. A 20-mer functional chaperone peptide (αB-crystallin peptide) derived from the amino acid residues 73–92 (DRFSVNLDVKHFSPEELKVK) of αB-crystallin protects RPE cells from OS-induced cell death by inhibiting caspase-3 activation [70,208].

The functions of ATP-independent chaperone HspB1 in RPE cells consist of blocking signaling pathways that trigger caspase-dependent apoptosis [209]. HspB1 is among the first chaperones to be activated in RPE cells under OS conditions, which decreases the level of ATP. These events are accompanied by the blocking of the external receptor-dependent pathway of cell death mediated by tumor necrosis factor receptors (TNFRs) and the internal mitochondrial signaling pathway. Then, after partial RPE cell recovery, ATP-dependent chaperone HSP70 is activated [209]. Hsp70 inhibits the formation of a functional apoptosome by interactions with Apaf-1. Hsp70 protects against the forced destructive action of caspase-3 and prevents the translocation of Bax from the cytoplasm to the mitochondria [210,211].

Hsp90 can prevent the formation of the apoptosome complex by inhibiting the oligomerization of Apaf-1 [210]. HSP90 regulates Akt activated by vascular endothelial growth factor (VEGF) and neuroprotectin D1 (NPD1) in the RPE in response to OS by inhibiting the signal cascade of dephosphorylation [212,213]. Akt is involved in the I3K-AktmTOR pathway, and its activation mainly occurs in the Nrf2-related response to OS and blocks autophagy. Proteasome inhibitors can enhance the autophagy process by inhibition of the PI3K–Akt–mTOR pathway, which makes it possible for them to be considered as therapeutic agents for enhancing anti-oxidative defense in human RPE cells [214,215,216].

The Hsp27 low-molecular-weight chaperone protein can maintain mitochondrial stability and redox homeostasis in cells and interacts with the apoptotic signaling pathways at many stages. Its activation leads to the blocking of Ca^2+^-induced apoptosis, which is a result of the suppression of caspase-3 functions and the prevention of cytochrome C release from Bcl-xS in the cytoplasm. HSP27 is also involved in the stabilization of Akt [217,218].

In addition to HSPs, other chaperones with anti-apoptotic functions are also known, for example, members of the Bcl-2-associated anthanogen (BAG) and clusterin (Clu) families [219,220]. Some of these proteins, such as BAGs likely serve to inhibit apoptosis by acting as co-chaperones for other proteins like HSPs, although there is recent evidence that BAGs may be cytoprotective by inducing autophagy [151,221].

### 7.2. Necroptosis (Regulated Necrosis)

Necrosis is activated by the production of proinflammatory factors, which leads to the destruction of the cell membrane. The necrotic pathway is initiated by the binding of TNF (tumor necrosis factor) to death receptors on the cell membrane, which is mainly controlled by RIP (receptor-interacting protein kinases) in the absence of caspase-8. Autophosphorylation of RIPK1 and RIPK3 leads to the formation of necrosomes, which are associated with mitochondrial dysfunction, leading to cell death [190,222,223]. In the RPE, the main characteristics of necrosis are ATP depletion and RIPK3 protein aggregation, cell swelling, and loss of cell membrane integrity under OS conditions [222,223].

Necroptosis is considered as regulated necrosis dependent on RIPK3 and MLKL (mixed lineage kinase domain) proteins. Caspase-8 activity can suppress this type of cell death by cleaving RIPK1 and RIPK3 [224]. The elimination of caspase-8 and FADD leads to autonomous activation of RIPK3 and MLKL that initiates the process of necroptosis [186,225]. It is interesting that RIP-mediated necroptosis becomes the predominant form of cell death after caspase inhibition. Therefore, necroptosis may serve as a backup mechanism along with apoptosis in various retinal diseases [226]. RPE cells respond to necrosis by enhanced producing of inflammatory cytokines that cause increased cell permeability. Necrosis-induced production of inflammatory cytokines in RPE cells have been demonstrated partially mimicked by recombinant HSP90 [227]. Inhibition of HSP90 by CDDO (synthetic triterpenoid, 2-cyano-3,12-dioxoolean-1,9-dien-28-oic acid) blocked necroptosis by inhibiting the activation of RIPK1 kinase [228].

### 7.3. Pyroptosis

Pyroptosis is an inflammatory form of cell death executed by gasdermins (GSDMs), a family of transmembrane pore-forming proteins activated via inflammasome-dependent or inflammasome-independent pathways [229].

Inflammasome activation is related to lysosomal destabilization. Inflammasomes contain the NOD-like receptor family pyrin domains, which are involved in initiating immune cell death by activating apoptotic or pyroptotic pathways in RPE cells [230,231,232]. Inflammasome priming by IL-1α, C5a, or medium conditioned by pyroptotic cells increased the susceptibility of RPE cells to photooxidative damage-mediated cell death and switched the mechanism of induced cell death from apoptosis to pyroptosis [56]. Two types of cell death, including pyroptosis and apoptosis, were activated in RPE cells after prolonged inflammasome activation induced by the drusen component of amyloid-beta (Aβ) [231]. All-trans retinal-derived A2E (*bis*-retinoid *N*-retinyl-*N*-retinylidene ethanolamine) of LF granules activates NLRP3 inflammasome to trigger pyroptosis or apoptosis of ARPE-19 cells [233].

Inflammasomes in the caspase-1-dependent pathway play an important role in triggering pyroptosis. The process is initiated by binding to proteins of the NLPR1, NLPR3, and NLPR4 (NOD receptor family) by inflammasomes, which activates caspase-1 with ASC as an adaptor protein. Caspase-1 promotes the cleavage of propyroptotic factor gasdermin D, producing an N-terminal fragment that induces cell death. Pyroptosis can be also activated independently of caspase-1. In this case, human caspase-4/5 and mouse caspase-11 promote the cleavage of gasdermin D to activate pyroptosis [191]. The NLRP3 inflammasome is a central regulator of inflammation and its activation has been associated with several age-related diseases, such as AMD, diabetic retinopathy, uveitis, and others [234]. 4-Hydroxynonenalis is one of the primary end products during LPO, which accumulates in RPE cells in AMD [235]. 4-HNE induces IL-6, IL-1β, and TNF-α production by promoting the extracellular efflux of HSP70 [236]. HSP70 affects inflammation by functioning as a negative regulator of the NLRP3 inflammasome [237].

HSP90 is a crucial chaperone protecting NLRP3 from destruction, keeping it intact but ready to be activated [238]. The inhibition of Hsp90 by TAS-116 (4-(1H-pyrazolo[3,4-b]pyridine-1-yl)benzamide) could prevent NLRP3 activation-dependent IL-1 release from RPE cells [239]. It is speculated that Hsp90 can assist in NLRP3 activation by stabilizing the P2X7 receptor [240]. Previously, it has been shown that Hsp90 inhibition can reduce NLRP3 inflammasome activation in RPE cells and that this effect relies on the activation of autophagy [241].

### 7.4. Ferroptosis

Ferroptosis is a cell death mode associated with LPO of polyunsaturated fatty acids leading to plasma membrane rupture. This form of death is characteristic of RPE cells due to their participation in the phagocytosis of POS, which constitute a major source of intracellular ROS and polyunsaturated fatty acids. The core metabolic mechanisms of ferroptosis are LPO and an imbalance of iron homeostasis [242]. Autophagy regulates ferroptosis by regulating cellular iron homeostasis and cell ROS production [243]. Ferroptosis is a major pathological process in OS-mediated RPE degeneration in cases of AMD, diabetic retinopathy, and others [244,245,246].

Recently, the involvement of chaperones in the induction and regulation of ferroptosis has been extensively reviewed [247]. The HSP90 family may act on GPX4 (glutathione peroxidase 4) and inhibit its antioxidant capacity [248]. The HSP90 family then participates in the regulation of ferroptosis through GSH/GPX4 pathway and inhibits LPO, therefore influencing ferroptosis [249]. HSP90 has also been identified as an important molecular chaperone that mediates the degradation of Gpx4 during ferroptosis, while suppressing ferroptosis in mouse neuronal HT-22 cells (*mouse hippocampal neuronal cell* line) [228]. A recent study has also demonstrated that overexpression of HSPA5 can negatively regulate ferroptosis by limiting Gpx4 degradation and LPO [250]. The upregulation of HSPA5 increases the expression and activity of GPX4, while GPX4 protects glioma cells from ferroptosis by neutralizing DHA-induced LPO [251]. The IP3R–HSPA9 (also known as GRP75)-voltage dependent anion channel 1 (VDAC1) complex bridges the gap in the mitochondria-associated membranes and establishes a platform for the transmission of ferroptosis signals from the ER to the mitochondria [252]. These findings establish a direct connection between the ER and the mitochondria mediated by calcium signals, with the ER acting as the initiator and the mitochondria as the effector of ferroptosis.

Overexpression of DNAJB6 (HSP40 family member) enhances the degradation of GSH, downregulates GPX4, enhances LPO, and promotes ferroptosis in esophageal squamous adenocarcinoma [247,253].

HO-1(HSP32), member of the sHSP family, has an anti-ferroptosis effect in human renal tubular epithelial cells, protecting AKI from ferroptosis by promoting GSH depletion [254]. Another member of this family HSPB1 also plays an integral role in ferroptosis. HSPB1 is considered to be a negative regulator of iron accumulation and uptake in fibroblasts or cardiac cells [227,255].

Under conditions of ferroptotic stress, there is an augmentation in the interaction between sigma non-opioid intracellular receptor 1 (SIGMAR1, also known as σ1R), a molecular chaperone situated in the mitochondria-associated membranes, and inositol 1,4,5-trisphosphate receptor (ITPR). This enhancement prompts an exchange of calcium ions between the ER and the mitochondria, thereby intensifying sensitivity to ferroptosis [252].

## 8. Extracellular Vesicles and Chaperones

RPE is a polarized epithelium that performs a barrier function, being located between the neural retina and the choroid. Extracellular vesicles (EVs) play an important role in the RPE interaction with adjacent tissues. They represent heterogeneous group of the extracellular particles that are delimited by a lipid bilayer and cannot replicate on their own. There are a number of EV subtypes (in particular, exosomes, which are a kind of small EV). EVs carry RNA, lipids, and proteins, such as heat shock proteins [256]. Exosomes released from the apical surface of the RPE cells contain αB-crystallin which may provide neuroprotection to neighboring photoreceptor cells [203]. Another function of αB-crystallin is inhibition of b-amyloid fibril formation. Minipeptides derived from αB-crystallin have been identified as anti-apoptotic agents in addition to their chaperone function [79]. αB-Crystallin is also a modulator of angiogenesis and vascular endothelial growth factor [79]. It is supposed that αB-crystallin controls the fusion of multivesicular bodies with the plasma membrane and the release of the exosomes [257].

In AMD, the secretion of RPE-derived EVs is enhanced to mediate OS, inflammation, angiogenesis, amyloid fibril formation, and drusen accumulation [258]. Extracellular microparticles enhanced senescence and interrupted phagocytic activity of RPE to lead to its degeneration [192]. In addition, stressed RPE cells released exosomes with a higher expression of VEGFR in the membrane and enclosed extra cargo of VEGFR mRNA. These exosomes may stimulate angiogenesis in choroids and during the healing of diabetic wounds [259]. Exosomes can also transport anti-inflammatory drugs to microglia, inhibit neuroinflammatory responses, and play a neuroprotective role in photoreceptor cells [260]. The cumulative evidence presented in recent review underscores the pivotal role of EVs in the onset and progression of retinal degenerative diseases (AMD, DR) [261].

## 9. Conclusions

The universal function of the RPE is the constant phagocytosis of the photoreceptor outer segment (POS) discs, which has led to the emergence of a unique form of phagocytosis without the formation of autophagosomes. The homeostatic balance of RPE cells is ensured by close interaction of endogenous defense systems indispensable to the health of the neural retina. Fine regulation of proteostasis and autophagy processes is primary to guarantee RPE homeostasis, protecting it from oxidative damage and protein accumulation. The redox balance in the RPE is largely dependent on autophagic clearance, which is a pleiotropic process by which cells deliver cytoplasmic components to the lysosome for degradation. Impaired autophagy significantly contributes into the development of RPE dysfunctions and photoreceptors degeneration in neural retina.

It is obvious that autophagy involves multiple mechanisms that use different pathways to maintaining cellular homeostasis, adaptation to stress, and regulation of immune responses and inflammatory processes. Analysis of scientific data demonstrates the multifaceted role of autophagy and much more specific endogenous selectivity of this process for the intracellular degradation and removal of destroyed and harmful components such as mitochondria (mitophagy), endoplasmic reticulum (ERphagy), peroxisomes (pexophagy), liposomes (lipophagy), and aggregated proteins (aggregophagy). Autophagy balance can act as an important switch between programmed and unprogrammed cell death.

Most of the chaperones are components of the molecular network of HSPs and their accessory proteins. HSPs promote cell survival through protection against changes in the cellular redox homeostasis, are implicated in junctional biogenesis, and can be responsible for the selective assembly of different junctional complexes. Recent studies have shown that various chaperones and co-chaperones are involved in the organization of the actin cytoskeleton, which is important for maintaining the stability of cell differentiation. HSPs can trigger or modulate multiple signals. HSP signaling pathways largely depend on the intensity of OS and on the state of RPE cells and demonstrates the possible use of alternative signal pathways.

The most of the molecular targets and signaling pathways of HSPs remain largely obscure. The extensive cross-talk between HSP signaling pathways and OS producers and mediators that control cellular response is problematic. This is due to the fact that HSPs integrate various mechanisms of endogenous defense systems into an overlapping molecular and genetic regulatory network and can perform oppositely directed functions. These proteins are involved in the regulation of proteostasis and maintenance of RPE cell viability, as well as in cell death signaling pathways. This makes it difficult to target the signaling to modulate autophagy processes and largely explains the controversial questions regarding insufficient selectivity and side effects of the HSPs activators or blockers for neuroprotection. Targeting HSPs may negatively affect other cell life support mechanisms and should be aimed at dissecting downstream signaling pathways. The multifaceted functions of molecular chaperones, HSPs, and their isoforms, co-chaperones, and HSP coinducers, as well as the spectrum of specific client proteins, at each step of proteostasis are still poorly understood. Understanding how HSPs in the RPE are regulated by both transcriptional and post-transcriptional mechanisms will promote the details of pathogenetic pathways of different forms of autophagy in RPE proteostasis in retinal disorders and shed light on potential strategies to treat visual impairments. The impact on molecular targets for autophagy by blocking the production of ROS or activating an endogenic defense requires knowledge on how to fine tune their participation in the regulation of the vital functions of RPE cells.

Inflammation and oxidative stress are thought to play major causative roles in the pathogenesis of many retinal degenerative diseases, such as age-related macular degeneration (AMD), diabetic retinopathy (DR), retinal vein occlusion, and retinitis pigmentosa [262,263]. This is evidenced by the abundance of clinical and experimental data. The current arsenal of strategies for neuroprotection uses antioxidant and anti-inflammatory therapies that are aimed at reducing oxidative stress and its consequences in RPE cells, which is usually accompanied by inflammatory processes, to prevent tissue damage.

Despite the common links in the pathogenesis of these diseases, they have their own specific development of inflammatory processes, which is largely associated with the intensity and duration of oxidative stress.

Alterations in autophagy processes are involved in the development of these ocular pathologies. In most ocular diseases (AMD, glaucoma, and cataract), proteolytic and autophagic capacity is attenuated, but excessive autophagic activity may accelerate the development of DR. Therefore, treatments focused on the modulation of autophagy processes in ocular RPE-related diseases could constitute a perspective therapeutic intervention [264,265]. It is important to emphasize that the effectiveness of autophagy-targeting drugs may have low specificity [266] since the targets are involved in multiple signaling pathways. In addition, the cross-talk between these pathways is not fully understood. Future studies using metabolomic approaches could identify and explore specific novel molecular targets that could pave the way to the prevention and targeted treatment of RPE-dependent retinal pathologies of specific diseases that lead to irreversible vision loss and blindness.

## Figures and Tables

**Figure 1 ijms-26-01193-f001:**
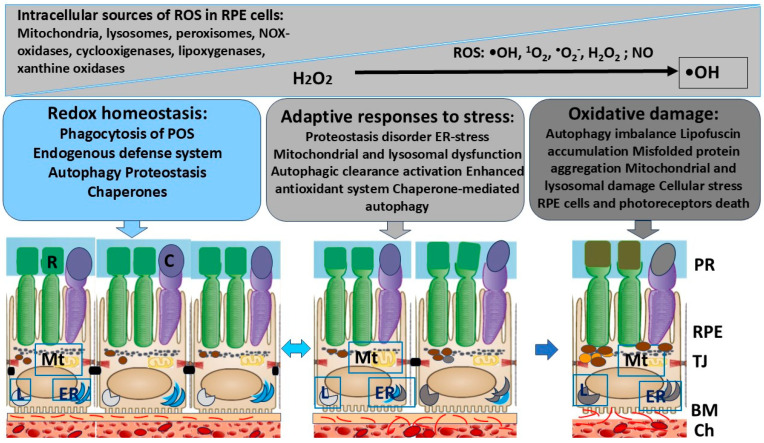
The main endogenous defense system that maintains RPE cell homeostasis. Given that the RPE is vital for key photoreceptor function, RPE dysfunction may lead to photoreceptor degeneration and severe visual impairment. RPE cells are characterized by a high metabolic activity and a predisposition for oxidative damage. The balance of RPE cell homeostasis is achieved through the coordinated work of endogenous regulatory systems, where the proteolysis and antioxidant protection systems play an important role. Disruption of redox homeostasis by endogenous and exogenous sources of ROS can lead to autophagy clearance activation, enhancement of the antioxidant system, chaperone-mediated autophagy activation, and, as a result, to an adaptive cell response. If these systems fail, oxidative stress can lead to the development of pathological processes and to the death of the RPE and photoreceptors. Abbreviations: TJ—tight junctions; BM—Bruch’s membrane; Ch—choroid; PR—photoreceptors; R—rod; C—cone; ER—endoplasmic reticulum; Mt—mitochondria; L—lysosomes; POS—photoreceptor outer segments; ROS—reactive oxygen species.

**Figure 2 ijms-26-01193-f002:**
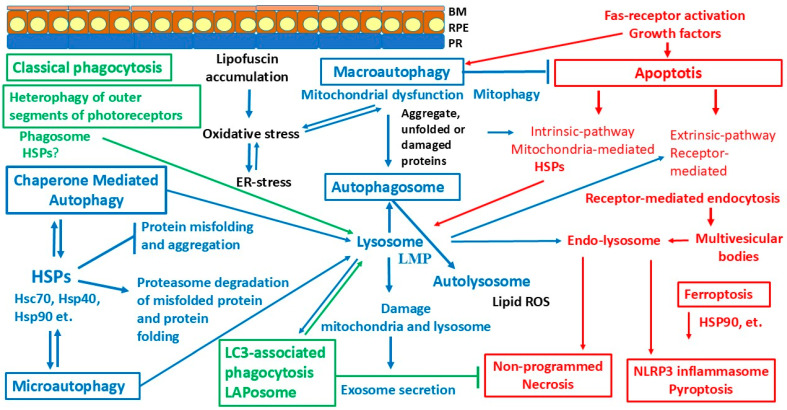
Schematic presentation of the intersection between degradative pathways in RPE cells. HSP involvement in phagocytosis, different types of autophagy, and cell death. RPE cells are constantly subjected to phagocytosis of photoreceptor outer segment discs and exposed to oxidative stress. Degradative pathways in RPE cells and associated processes are indicated by arrows with the corresponding colors: types of phagocytosis (green), types of autophagy (blue), and forms of cell death (red). Photoreceptor outer segment discs are phagocytosed by the RPE (heterophagy) and are degraded in the lysosomes in the process of LAP. LAP bridges the phagocytic and canonical autophagic pathways. Abbreviations: RPE—retinal pigment epithelium; ROS—reactive oxygen species; ER—endoplasmic reticulum; LMP—lysosome membrane permeabilization; LAP—LC3-associated phagocytosis; HSPs—heat shock proteins.

**Table 1 ijms-26-01193-t001:** Main features of major types of programmed cell death (modified from [184,192,193,194,195]).

Characteristics		Apoptosis	Necroptosis	Ferroptosis	Pyroptosis
Morphology	Cell	Shrinkage	Swelling	Rounding up	Bubbling
	Plasmatic membrane	Blebbing	Pore formation	Lack of rupture and blebbing	Rupture
	Subcellular structures	Pseudopod retraction	Necrotizing bodies	Rupture of outer mitochondrial membrane	Pyroptotic bodies
	Nuclei	Fragmentation	Moderate chromatin condensation	Lack of condensation	Chromatin condensation
Activated/increased markers		Caspase3/7 Cytochrome C release	RIPK1,3 MLKL PS exposure	Iron and ROS accumulation Lipid peroxidation MAP kinases	NLRP3 ASC Pro-caspase-1 Gasdermin D
Outcome		Engulf cells by phagocytes without inflammation	Cell lysis Inflammation	Inflammation	Cell lysis Inflammation

Abbreviations: ASC—apoptosis-associated speck-like protein containing a CARD; MLKL—pseudokinase mixed lineage kinase-like; NLRP3—inflammasome; PS—phosphatidylserine; RIPK—receptor-interacting kinase.

## Data Availability

Not applicable.

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
