# Peer review of "Retinal Pigment Epithelium Under Oxidative Stress: Chaperoning Autophagy and Beyond"

_ijms, 2025, doi:10.3390/ijms26031193_

Round 1
Reviewer 1 Report
Comments and Suggestions for Authors
The manuscript entitled Retinal Pigment Epithelium Under Oxidative Stress: Chaperoning Autophagy and Beyond brings a new inside in the field of oxidative stress and inflammation effects on retina.
Observations
line - 38-40 - reformulate this sentence because is difficult to understand your idea.
line 40-43 - the same observation
line 44 - toissue or structure? Tissues are structures. You can replace it with tissue's structure if this is what you mean.
line 76 - what kind og review do you performed? Plese write details about the material and method.
line 81 - what do you mean by ocular system??
line 84-88 - I suggest to make a table with RPE functions to be more clear.
Fig. 2 - what different colors represent in this figure? This must be specified in the legent and afterwards in the text.
Conclusions
Erase line 774-778 paragraph. This is the aim of the study and you should not put in the Conclusions chapter. Plese focus on the main idea of your work.
I suggest that all the oxidative stress and inflammation mechanisms you describe in your manuscript to be emphasize like contributors to retoina disease. Please name it and eventually organize it in a table as up to date about their importance in pathophysiological mechanism involved in retinal disorders. This mechanism are very important for new therapies addressed to retinal pathology.
Comments on the Quality of English Language
The manuscript needs a professional English help.
Author Response
Comment 1 line - 38-40 - reformulate this sentence because is difficult to understand your idea. Response 1 We agree with this comment. Therefore, we have reformulated this sentence (page 1, lines 38-40) Comment 2 line 40-43 - the same observation Response 2 Agree. It was reformulated (page 1, lines 40-43) Comment 3 line 44 - tissue or structure? Tissues are structures. You can replace it with tissue's structure if this is what you mean. Response 3 Replace it with tissue's structures (page 1, lines 44-45) Comment 4 line 76 - what kind og review do you performed? Please write details about the material and method. Response 4 Agree. We clarified our approach to the selection of material for our review (page 2, lines 73-78) Comment 5 line 81 - what do you mean by ocular system?? Response 5 We changed "ocular system" to "these tissues" (page 2, line 83) Comment 6 line 84-88 - I suggest to make a table with RPE functions to be more clear. Response 6 When compiling a table of PE functions, we were faced with the need to determine what is a function and what is a process. In general, it is natural to think of a process as answering the “how” question and a function as answering the “for what” question. However, depending on the context, the same thing can be considered either a function or a process. Thus, in PE, phagocytosis is presented as a process consisting of several stages, but at the same time providing one of the most important functions - the utilization of the outer disks of photoreceptors. This duality and context-dependency of process-function definition is especially clearly demonstrated in the following reviews: Xie et al., 2016: Ferroptosis: process and function. Cell Death Differ. 23(3):369-79 Mizushima, 2007: Autophagy: process and function. Genes Dev. 2007. 21(22):2861-73 Taking into account the above, we limited ourselves to introducing some clarifications into our description of the functions of RPE (page 2, paragraph 2, lines 87-94) Comment 7 Fig. 2 - what different colors represent in this figure? This must be specified in the legent and afterwards in the text. Response 7 We specified in the legent of figure 2 what different colors represent (page 5, lines 191-192) Comment 8 Conclusions. Erase line 774-778 paragraph. This is the aim of the study and you should not put in the Conclusions chapter. Plese focus on the main idea of your work. Response 8 We agree with this comment and erase these lines (page 17, paragraph 9) Comment 9 I suggest that all the oxidative stress and inflammation mechanisms you describe in your manuscript to be emphasize like contributors to retina disease. Please name it and eventually organize it in a table as up to date about their importance in pathophysiological mechanism involved in retinal disorders. This mechanism are very important for new therapies addressed to retinal pathology. Response 8 We agree with this comment. Unfortunately, the presence of numerous variants for hereditary eye diseases associated with mutations of various genes (as, for example, in the case of retinitis pigmentosa) did not allow us to present the data concerning the mechanisms of OS and inflammation in the form of a table. However, given the importance of these data for therapy, we added a brief comment on the role of these processes in hereditary retinal diseases associated with RPE degeneration (page 18, paragraph 9, lines 833-855)Reviewer 2 Report
Comments and Suggestions for Authors
This review delves into the intricate mechanisms of autophagy within retinal pigment epithelium (RPE) cells when subjected to oxidative stress. It offers a comprehensive overview that can inform strategies for preventing and treating retinal degenerative diseases.
Recent studies (Cell,2019 177, 428–445; J Extracell Vesicles. 2024;13:e12404) have highlighted the enrichment of heat shock proteins (HSPs) within extracellular vesicles (EVs). Moreover, the literature (Interdiscip. Med. 2023;1:e20230019) highlighted that oxidative stress triggers RPE cells to release EVs, upregulating vascular endothelial growth factor receptors (VEGFRs) within these vesicles and subsequently influencing angiogenesis. Based on these findings, we hypothesize that RPE cells under oxidative stress may utilize EVs as a means of intercellular communication to modulate retinal function.
Therefore, we propose that this review should consider the role of EVs in the context of RPE autophagy and oxidative stress. A deeper understanding of EV-mediated signaling pathways in the retina could provide novel insights into the pathogenesis of retinal diseases and identify potential therapeutic targets.
Author Response
Comment 1. Therefore, we propose that this review should consider the role of EVs in the context of RPE autophagy and oxidative stress. A deeper understanding of EV-mediated signaling pathways in the retina could provide novel insights into the pathogenesis of retinal diseases and identify potential therapeutic targets. Response 1 We agree with this comment. Therefore, we have added special paragraph dedicated to the role of Evs in the RPE with focus to HSPs (pages 16-17, paragraph 8)Reviewer 3 Report
Comments and Suggestions for Authors
Some issues need to be addressed before paper went to the next step. As follows:
1. Line 47, it should be [12, 13]
2. I suggest modifying Figure 1. There are too many elements and colors, making it look chaotic.
3. Please consider to merge 6.3 and 6.4 parts into one part.
4. In the section 7, I suggested author provides a figure that shows the similarities and differences in different cell death patterns in response to oxidative stress.
Author Response
Comment 1 Line 47, it should be [12, 13]
Response 1 Corrected (page 1, line 46)
Comment 2 I suggest modifying Figure 1. There are too many elements and colors, making it look chaotic.
Response 2 We agree with this comment and modify Figure 1.
Comment 3 Please consider to merge 6.3 and 6.4 parts into one part.
Response 3 We don't find possible to merge these paragraphs. Paragraph 6.3 (LC3-associated phagocytosis) is considering one of type of phagocytosis, and may be merge more with paragraph 6.2 than 6.4 (Autophagy).
Comment 4 In the section 7, I suggested author provides a figure that shows the similarities and differences in different cell death patterns in response to oxidative stress.
Response 4 We agree with this comment. We provided a table (Table 1) which summarizes main features of major types of programmed cell death (page 13, paragraph 7, lines 594-596)
Round 2
Reviewer 2 Report
Comments and Suggestions for Authors
Accepted